# Fleeing the war: A socio-ecological perspective on the mental health of internally displaced and refugee children and adolescents living in the Kurdistan region of Iraq

**Jasmin Wittmann** [1]*, **Hawkar Ibrahim** [1,2], **Frank Neuner** [1,2], **Claudia Catani** [1,2]

**1** Department of Psychology, Clinical Psychology and Psychotherapy, Bielefeld University, Bielefeld, Germany, **2** Vivo International, Konstanz, Germany

\* jasmin.wittmann@uni-bielefeld.de

**Data Availability Statement:** The datasets generated and analyzed in the current study are not

## Abstract

Over the past decade, the number of children forced to flee their homes worldwide has increased twofold. The accumulative experiences of war, displacement, and flight can have a devastating impact on the mental health of affected minors. Although more than half of all displaced minors are internally displaced within their countries of origin, little is known about the psychopathology of these displaced children living in regions of ongoing or recent conflict. Employing a socio-ecological perspective, this study aims to identify risk factors contributing to psychopathology among internally displaced and refugee minors living in camps for displaced families in the Kurdistan region of Iraq. A total of 332 displaced children and adolescents, aged between 8 and 16 years, were interviewed by trained paraprofessionals in a cross-sectional study in 2019. Trauma and mental health symptoms, including posttraumatic stress disorder, depression, and internalizing and externalizing problems, were assessed. The findings highlighted elevated rates of trauma exposure and psychopathology among the participating minors. Using multiple linear regression analyses we identified risk factors across all mental health outcomes at the individual level (e.g., higher age, greater trauma exposure) and the family level (e.g., greater family violence). Moreover, an analysis at the community level, dividing the study sample by legal status (internally displaced vs. refugee) revealed significantly higher rates of trauma exposure and psychopathology among internally displaced minors. These findings have implications for developing appropriate support for the healthy development of forcibly displaced children and adolescents living in areas of ongoing conflict. Prevention and intervention strategies should take into account various socio-ecological levels, including trauma-focused psychotherapy at the individual level, measures to prevent violence at the family level, and community-level actions that consider context-specific responses, such as adapting camp conditions to meet the needs of vulnerable displaced children.

publicly available due to the terms of consent agreed to by the participants. Further, the datasets contain sensitive data (collected through in-depth clinical interviews in a vulnerable population) with indirect identifiers (such as age, sex, nationality, legal status, location, history of traumatic events) that might risk the identification of study participants. However, a minimal dataset will be available upon reasonable request either from the corresponding author or from the data custodian at the Department of Clinical Psychology, Bielefeld University, Dr. Benjamin Iffland (benjamin. iffland@uni-bielefeld.de). In the long term, the data will be stored in encrypted form on the Bielefeld University's internal hard drive, accessible to the authors and the departmental data custodian. The terms of consent obtained from the participants as well as the data protection procedure are in accordance with the ethical guidelines of the Research Ethics Committee of the Faculty of Science and Health, Koya University, Iraq, and the Ethics Committee of the University of Bielefeld, Germany, which provided ethical approval for the study.

**Funding:** This study was funded by the Volkswagen Foundation (Grant number 91474-1). The funder had no role in study design, data collection, data analysis, decision to publish, or preparation of the manuscript.

**Competing interests:** The authors have declared that no competing interests exist.

## Introduction

The number of minors uprooted from their homes has been increasing in the last decade. According to the latest report by the United Nations International Children's Emergency Fund [1] on the state of the world's children, at the end of 2021, about 36.6 million children and adolescents had been forcibly displaced; amounting to 41% of all displaced persons. Forcibly displaced persons are exposed to traumatic events–such as armed conflict, violent destruction of their homes or loss of a family member–in their place of origin, while fleeing and in their place of resettlement. This elevated trauma exposure puts displaced persons at an increased risk of mental health problems. Children and adolescents are particularly vulnerable to the devastating mental health effects of violent conflict and persecution as they are less able to protect their interests before, during and after natural and man-made disasters [2]. In addition, uncertainty and lack of security at multiple levels interfere with crucial formative stages of healthy child development [3–5]. Consequently, high prevalence rates of psychopathology such as posttraumatic stress disorder (PTSD) as well as internalizing (e.g. anxiety or depression) and externalizing problems (e.g. risky or aggressive behavior) have been found among displaced children and youth [6–10]. A recent meta-analysis on the mental health of displaced minors [11] also revealed high overall rates of PTSD (22.71%) and depression (13.81%). These mental strains are usually accompanied by functional impairments in daily life and have harmful effects on a children's development by increasing the risk of permanent disability [12–14]. Even though the majority of displaced populations seek refuge either in their country of origin or in a neighboring country, knowledge on the mental health of minors residing in areas of ongoing or recent conflict is particularly scarce [3, 11, 15–17]. In order to promote healthy child development in this vulnerable population, it is important to understand the specific contribution of various risk and protective factors on the different aspects of child psychopathology. In (post-)war and conflict settings, children and adolescents face adversity at multiple and intersecting socio-ecological levels, such that an interplay of individual, family and community risk and protective factors influence their mental health [3, 18–20].

At the individual level, cumulative trauma exposure has proven a major risk factor for displaced children's mental health across various settings. Recent reviews have found that exposure to more types of traumatic experiences were associated with higher levels of psychological distress including PTSD, depression, anxiety and externalizing problems [3, 8, 21–24]. Relationships between gender, age and psychopathology have also been explored. While the current state of research on the association between gender and the development of psychopathology in traumatized individuals is mixed [25, 26], psychopathology in war-exposed children seems to increase with age [27].

At the family level, mass trauma contributes to an intergenerational cycle of violence. Previous research has shown links between parental factors (e.g. parental exposure to war trauma, maternal revictimization, parental psychopathology) and an increased vulnerability to dysfunctional parenting [28–31]. Consequently, being raised in (post-)conflict regions, children have a higher risk of experiencing child maltreatment, which has consistently been identified as a key determinant for a minor's mental health [21, 31–35]. Most importantly, studies in post-conflict settings have shown that both the experience of family violence as well as war trauma made unique contributions to different aspects of child mental health such as PTSD [34], depression [36, 37], and internalizing and externalizing behavior problems [19]. However, most of the cited studies have been conducted in Central and Eastern Africa (Rwanda, Uganda, Tanzania) and Central and South Asia (Afghanistan, Sri Lanka), while there is a lack of data on the differential contribution of various childhood adversities on the mental health of children in the Middle East and North Africa (MENA) region. Marked by protracted and

high-intensity conflicts over the past decade, over a third of children in the MENA region are affected by ongoing conflict and violence, including 13.3 million forcibly displaced minors [38], highlighting the need for empirical evidence from this region. Home to a significant proportion of the world's displaced population, forced migration in the MENA region is mainly characterized by cross-border movements of refugees (15%) and internally displaced populations (IDPs, 78%) seeking protection in their country of origin [39]. Although to date, IDPs account for 57.7% of the world's displaced population [40], evidence on the mental health of this specific group is still scarce [16, 17]. Thus, when looking at displaced war survivors in ongoing and post-conflict regions such as MENA, a comparison of the two groups of internally displaced and refugee children could shed light on the role of one's legal status as a potential determinant of mental health at the community level. The legal status of a displaced person has different implications in terms of protection and assistance; while refugees are covered by international law, which entitles them, for example, to humanitarian aid and international financial assistance, IDPs remain under the protection of their own governments, which puts them in a particularly vulnerable position. Consistent with this line of thinking, initial evidence among displaced adults suggests that mental health outcomes are worse among IDPs compared to refugees [17, 41, 42], however, related research in displaced minors in regions of ongoing or recent conflict is still scarce. A recent systematic review of the mental health of refugee children and adolescents seeking protection in high income countries identified an insecure legal status, including uncertainty about the future and potential deportation to their place of origin, as risk factors for poor mental health [8]. In a longitudinal cohort study in asylum seeking children and adolescents in Germany, minors who had received a rejection of their asylum application presented with significantly more psychological distress in the one year follow-up interview [43].

Therefore, when adopting a socio-ecological model to conceptualize different risk factors for adverse mental health outcomes among children in (post-)conflict settings, in addition to factors at the individual and family level (trauma exposure, family violence), legal status (e.g., IDP vs. refugee) should be considered as a community-level determinant that may influence the psychopathology of affected minors. Against this background, the present study focuses on the mental health of displaced children and adolescents, both IDPs and refugees from Iraq and Syria living in a camp in the Kurdistan Region of Iraq (KRI). Since the Arab Spring starting in 2010, the Middle East region has experienced significant turmoil. In Iraq and Syria, the ongoing civil war has included a humanitarian crisis that has forced millions of Iraqis and half of the Syrian population to flee (UNHCR, 2018). The majority of refugees and IDPs in Iraq found refuge in the KRI [44]; the setting of the present study. Previous research suggests high levels of psychopathology in both populations. For instance a study on the mental health of Syrian refugee children in a camp in Jordan suggests high levels of anxiety and depressive symptoms [45]. Among internally displaced children in Iraqi Kurdistan living in displacement camps, a PTSD rate of 87% was documented five years after the experiencing a chemical attack [46]. The present paper aims at examining the mental health and related risk factors at different socio-ecological levels (individual level: exposure to war and adverse events, family level: exposure to family violence, and community level: legal status as an IDP or refugee) in displaced minors living in a region of ongoing conflict, the KRI. In a heterogeneous sample of refugees as well as internally displaced children and adolescents with varying degrees of war and family violence, we were able to examine the distinct contribution of different child adversities to various aspects of psychopathology (PTSD, depression, and internalizing and externalizing problems). We hypothesized that both the amount of war adversities and family violence would have a specific negative effect on all aspects of mental health in displaced minors. A unique feature of our study is the participation of both refugee children and adolescents (RC)

as well as internally displaced children and adolescents (IDC). As the generalizability of trauma-related psychopathology from one displaced group to another is limited, even when sociodemographic characteristics, time of flight and place of resettlement are similar [47], we aimed to identify the specific needs of the two groups of displaced minors. By examining the study sample separately according to their legal status, we aimed to answer the following questions: Are mental health problems higher among IDC compared to RC, and do the same risk and protective factors contribute to the mental health of forcibly displaced IDC and RC?

## Materials and methods

### Ethics statement

Ethical approval for the study was obtained from the ethical committees of Bielefeld University, Germany (reference number: EUB 2015–046) and Koya University, KRI (reference number: SHETC-1). The study and its procedure was also approved by local government departments, including the protection office in Joint Crisis Coordination Centre (JCC) in the Ministry of Interior and Directorate of Social Affairs (DoSA) in the Ministry of Labour and Social Affairs. Recruitment of the participants was based on the principle of informed consent which was obtained from their parents as well as their own agreement. Consent was given verbally and documented by the interviewers. We had to refrain from written consent because of potential legal consequences for the participants [48].

### Participants and procedure

The study procedure, sampling plan, inclusion criteria for participants, data collection quality assurance, and detailed ethical considerations are described elsewhere [48, 49]. From 2 March to 29 April 2019, a sample of 322 children and adolescents (51.8% female, 48.2% male) from Syria and Iraq were interviewed by trained local bachelor-level psychologists and social workers in Arbat Camp, Sulaymaniyah Governate in the Kurdistan Region of Iraq (KRI). Arbat camp is hosting displaced persons of different nationalities (Syrian and Iraqi), ethnicities (Arabs and Kurds) and religions (Muslims and Yazidis). In the final study sample, children and adolescents ranged in age from 8 to 16 years. Approximately half of the children and adolescents were of Kurdish ethnicity and were interviewed in Kurdish (54.8%) and half were of Arab ethnicity and were interviewed in Arabic (45.2%). While the key instruments had previously been developed and validated specifically for the study context in Arabic and Kurdish as described below [48, 50], the remaining instruments were assessed using our own translations. Trained interviewers fluent in the local languages adapted the wording according to the specific individual language skills and preferences of each responding minor. Table 1 provides an overview of the socio-demographic characteristics of the sample as a whole and of the two study sub-groups, IDC and RC, separately.

### Instruments

**Sociodemographic information.** First, children and adolescents were asked about their sociodemographic information such as age, gender, nationality, ethnicity, education and family characteristics, followed by questions about characteristics of their life before and after the flight as well as during the war.

**Traumatic experiences and family violence.** Children and adolescents' exposure to war-related traumatic events and adversities was measured by the War and Adversity Exposure Checklist [50]. Created specifically for the Middle East population, the checklist consists of 26 items asking about war-related events and other traumatic experiences in a two-response

**Table 1. Sociodemographic characteristics and traumatic experiences.**

| | Total sample | Study subgroup (based on legal status) | | |
| --- | --- | --- | --- | --- |
| | (N = 332) | IDC (N = 174) | RC (N = 158) | p |
| Age, Mean (SD)[a] | 12.67 (2.08) | 12.51 (2.11) | 12.85 (2.14) | - |
| Gender–female | 51.8% | 49.4% | 54.4% | - |
| Ethnicity | | | | |
| Arab | 45.2% | 86.2% | - | *** |
| Kurd | 54.8% | 13.8% | 100% | |
| Religion | | | | |
| Muslim-Sunni | 96.4% | 94.3% | 98.7% | ** |
| Yezidi | 3.0% | 5.7% | - | |
| Atheist | 0.6% | - | 1.3% | |
| Education, Mean (SD)[a] | 4.96 (2.17) | 4.83 (2.19) | 5.10 (2.14) | - |
| Number of siblings, Mean (SD) | 5.45 (3.07) | 6.73 (3.07) | 4.04 (1.94) | *** |
| Growing up in | | | | |
| City or town | 58.1% | 31.6% | 87.3% | *** |
| Village | 41.9% | 68.4% | 12.7% | |
| Age during war, Mean (SD)[a] | 7.34 (2.39) | 7.48 (2.26) | 7.19 (2.52) | - |
| Place of living during war | | | | |
| Urban areas | 55.5% | 33.9% | 79.1% | *** |
| Rural areas | 44.6% | 66.1% | 20.9% | |
| Traumatic experiences | | | | |
| War and adversities, Mean (SD) | 4.45 (3.84) | 5.41 (4.04) | 3.39 (3.31) | *** |
| Family violence, Mean (SD) | 1.73 (2.71) | 2.25 (3.24) | 1.16 (1.80) | *** |

Note.

*indicate significant differences between the two subgroups on

**p≤.01

***p≤.001

[a] in years

format (yes/no). An index of exposure to war related trauma was established by summing up all reported types of violence on the checklist. To assess exposure to family violence, we used the short version of a checklist that had been created to investigate adverse experiences in the family context during childhood and was previously used in several post-conflict contexts such as Northern Sri Lanka and Uganda [28, 29, 51]. The questionnaire consisted of 13 items assessing different types of physical, emotional and sexual abuse as well as neglect, and two items about potential consequences of the experienced family violence. For each event, the children and adolescents were asked if they had experienced it or not (yes/no). Again, an index of family violence was established by aggregating all the event types on the checklist reported by the child.

**PTSD symptoms.** PTSD was assessed using the Posttraumatic Stress Interview for Children (KID-PIN) [48], a semi-structured interview for PTSD symptoms. The instrument has recently been developed as a contextually appropriate and psychometrically sound screening tool for PTSD in children and adolescents in the Middle East that can be administered by both clinicians and trained paraprofessionals. For 20 symptoms, children and adolescents had to rate the frequency over the past month on a three-point scale (0–2) and severity on a five-point scale (0–4), if the symptom occurred more than once. A PTSD score is obtained by summing up the scores of each item. The instrument showed excellent internal consistency (Cronbach's alpha = .94).

**Depression symptoms.** The short version of the Mood and Feelings Questionnaire–Child Version [52] was used to examine symptoms of depression. The 13 items on the scale are measured on a three-point Likert scale (0–2), and depression scores, which range from 0 to 26, are obtained by summing the item scores. The internal consistency was $\alpha = .89$.

**Emotional and behavioral problems.** The Strengths and Difficulties Questionnaire SDQ [53] was used to assess internalizing and externalizing behavior problems. Being widely used, the measure shows good psychometric properties [54]. A total of 25 items cover the five subscales: emotional problems, peer problems, conduct problems, hyperactivity and prosocial behavior. An internalizing problem score (0–20) is obtained by combining the sum of the emotional problems and peer problems scales, while the scores of the conduct problems and hyperactivity scales add up to an externalizing problem score (0–20). The total difficulties score (0–40) is obtained by combining the internalizing and externalizing problem scores. Comparable to other studies [19], the internal consistency of the total difficulties score was $\alpha = 0.63$.

## Statistical analysis

Data entry and statistical analysis was carried out using IBM SPSS Statistics (Version 25) and data was visualized using Microsoft Excel. Descriptive statistics were used for the description of the general characteristics of the sample, the traumatic events and the mental health symptoms. The normality assumption was checked with Kolmogorov-Smirnov and Shapiro-Wilk tests as well as by visual inspection of histograms and Q-Q-Plots. Group differences were analyzed with two-tailed independent sample t-tests for normally distributed variables, Mann-Whitney U tests were conducted for non-normally distributed variables and Chi-square tests of independence were performed to examine group differences for nominally categorized variables. We tested correlations between the continuous variables with Spearman-Rank correlations, and point-biserial correlations for dichotomous variables. Differences between the study subgroups based on legal status (IDC and RC) were tested for all psychopathology measures (PTSD, depression, and emotional and behavioral difficulties). To examine potential predictors for children's psychopathology we conducted multiple linear regression analyses for the whole sample and separately for the two study subgroups. For each of the outcomes, the following variables were included as predictors: age, gender, number of siblings, number of different traumatic event types and amount of family violence. We have selected the predictors based on the existing literature guiding our theoretical framework as well as our assessment of the interpretability of the variables. The following variables were not included in the regression models for the following reasons: *Level of education*: During data collection, we found that the interpretability of this variable (number of years of education) was limited. As there is no comprehensive, functioning school system in the camps, this variable is more related to factors such as age and gender, rather than reflecting the actual education level of the study sample. *Religion and ethnicity*: There is insufficient empirical evidence to support the hypothesis that mere religious or ethnic affiliation is independently associated with elevated mental health symptomatology. Consequently, we have refrained from positing such associations in our analysis. *Place of residence during the war*, *place of residence before the war*: We excluded these variables because they rely on retrospective information from several years ago. This does not reflect the children's current living conditions, as they had been in the camp for an average of four years. As the residuals for the outcome variables PTSD and depression deviated significantly from homoscedasticity, a 1.000 samples bootstrapping procedure, as a widely accepted method for addressing heteroscedasticity in multiple linear regression models [55] was applied for these variables. The regression models fulfilled all other necessary quality criteria for linear

regression analysis. Furthermore, item number 17 of the WAEC checklist (*physical assault by family member*) was excluded for the calculation of the traumatic events score that was used in the regression analyses, as this content is also covered by the family violence checklist. A p-value of $<0.05$ was considered to be significant.

## Results

### Exposure to war related events and adversities

Participants reported exposure to several traumatic and adverse events. While only 8.1% (n = 27) of the sample did not endorse any traumatic event, more than half of the sample (60.8%) had experienced three or more different types of traumatic events. In total, the children and adolescents experienced between 0 and 20 different types of traumatic events (*M* = 4.45, *SD* = 3.84). 74.7% of the IDC were exposed to three or more traumatic event types, compared to 48.0% of the RC. As indicated in Table 1, on average, IDC reported significantly more exposure to war related events and adversities, compared to the RC. Fig 1 illustrates the most common experiences of war-related events and adversities in the sample.

### Exposure to family violence

55.4% (n = 174) of the children and adolescents reported at least one event of family violence. The children and adolescents had experienced between 0 and 15 different types of events on the family violence spectrum. The most commonly reported experiences were being shouted at or sworn at and being hit or beaten up. Again, as presented in Table 1, in comparison to RC, IDC reported experiencing significantly higher levels of family violence.

### Mental health symptomatology

PTSD. Using a cut-off score of 28 [48], the prevalence of a diagnosis of probable PTSD was 21.4% across the whole sample (*M* = 13.34, *SD* = 18.02). Nearly a third of the IDC (27.6%) scored above the cut-off for probable PTSD, compared to 14.6% of the RC. Mean scores of children and adolescents' psychopathological symptoms for the whole sample as well as for the two sub-groups are shown in Fig 2.

Depression. Using a cutoff score of 10 [56], 14.2% of all children and adolescents (*M* = 3.69, *SD* = 5.17) evidenced probable depression. 16.1% of IDC scored above the cut-off, compared to 12.0% in the RC.

Emotional and behavioral difficulties. Using the cutoffs proposed by Goodman [53] (a total SDQ-score of 0–15 indicating low, 16–19 indicating medium, and 20–40 indicating high risk for mental disorders), the SDQ identified 7.5% of the children and adolescents (IDC: 10.9%, RC: 3.8%) having a high risk for mental disorders and another 7.5% of the children and adolescents (IDC: 6.3%, RC: 8.9%) with a medium risk for mental disorders according to the children and adolescents' self-report (total sample: *M* = 10.22, *SD* = 5.29, IDC: *M* = 10.61, *SD* = 5.60, RC: *M* = 9.80, *SD* = 4.91).

### Predictors of mental health symptoms

The details of the multiple linear regression analyses that were carried out for the whole sample and separately for the two study groups (RC and IDC) are presented in Tables 2–4.

Total sample (Table 2). Results from all models showed that older age, greater exposure to war-related traumatic events and experiences of family violence were positively associated with mental health symptoms. Furthermore, being female was a significant predictor for the severity of depression symptoms.

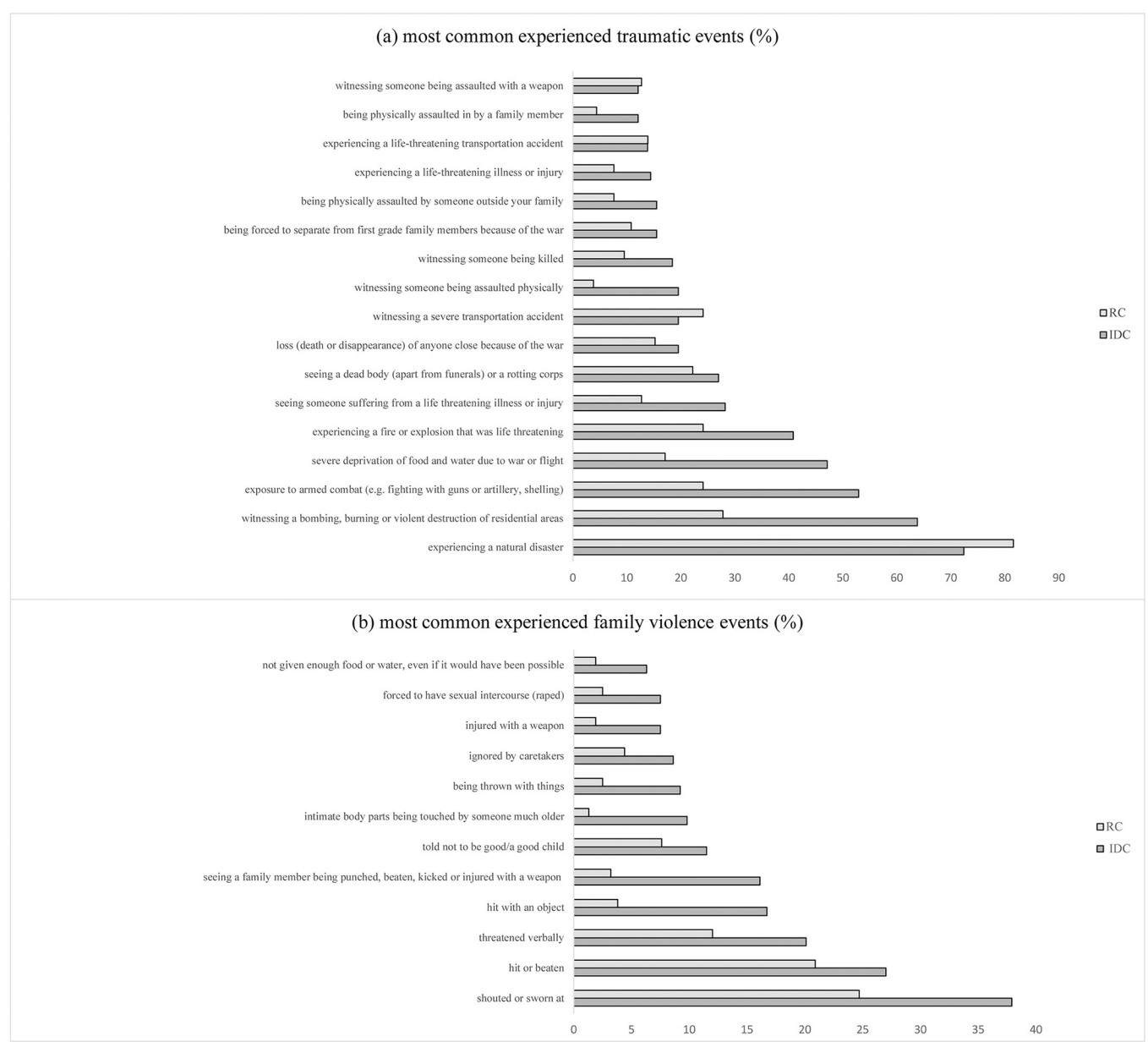

**Fig 1.** Percentages of children and adolescents exposed to (a) events of war and adversities and (b) incidents of violence at home.

RC (Table 3). Exposure to war-related and adverse events emerged as a significant predictor of all mental health problems in children and adolescents with refugee status. In addition, exposure to family violence was significantly associated with all outcomes except for depression symptoms. Furthermore, older age and being female were significantly associated with depression symptoms, being female was also significantly related with externalizing problems.

IDC (Table 4). Exposure to war-related and adverse events was a significant predictor for all outcome measures, while experiences of family violence emerged as a significant factor for externalizing problems. In addition, older age was significantly related to PTSD and depression symptoms and the number of siblings was negatively associated with depression symptoms.

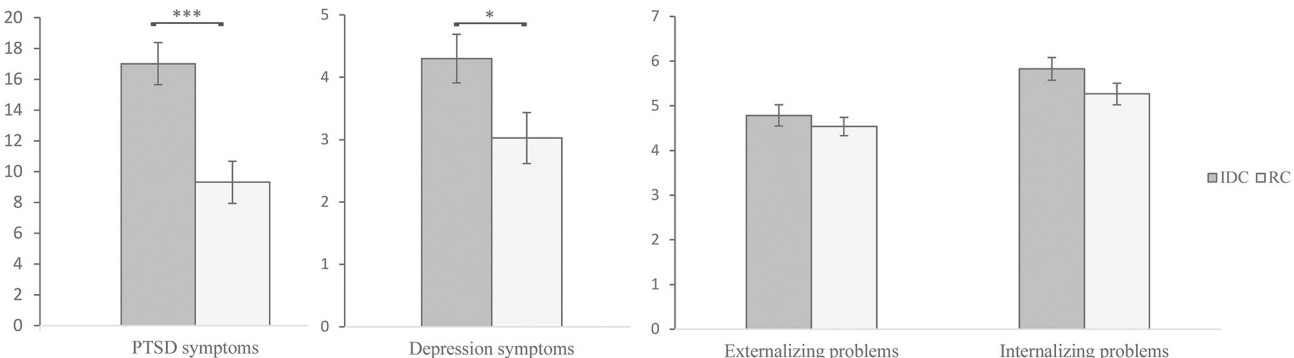

**Fig 2. Mean scores of children and adolescents' psychopathological symptoms and statistical differences between the two sub-groups.** PTSD Symptoms: KID-PIN, Depression symptoms: SMFQ, Externalizing and internalizing problems: SDQ. Standard errors are presented as error bars, ***p≤.001, *p≤.05.

## Discussion

The current study was able to show that the mental health of forcibly displaced children and adolescents in the aftermath of armed conflict is shaped by an interplay of risk factors on different socio-ecological levels, such as gender, age, war trauma and family violence. Overall, internally displaced children and adolescents showed a higher burden of trauma and mental health problems compared to children and adolescents with refugee status.

The region's ongoing decades-long conflict is reflected in the high number of children and adolescents (91.9%) reporting traumatic experiences related to war and conflict. The level of mental health impairments in the present sample was exceptionally high compared to displaced minors resettled in high-income countries (see 7 for a review). Thus, our data suggest

**Table 2. Results of linear regression models on the different types of child psychopathological symptoms–total sample (N = 322).**

| Predictor | PTSD symptoms[a] | | Depression symptoms[b] | | Internalizing problems[c] | | Externalizing problems[d] | |
|---|---|---|---|---|---|---|---|---|
| | Standardized β-Coefficient | zero-order correlation(ρ) | Standardized β-Coefficient | zero-order correlation (ρ) | Standardized β-Coefficient | zero-order correlation (ρ) | Standardized β-Coefficient | zero-order correlation (ρ) |
| Age | 0.13** | 0.12* | 0.23*** | 0.18** | 0.11* | 0.11* | 0.11* | 0.11* |
| Gender (male)[e] | -0.08 | -0.07 | -0.14*** | -0.16** | -0.07 | -0.08 | 0.01 | 0.00 |
| Legal Status (Syrian RC)[e] | -0.08 | -0.30*** | -0.04 | -0.21*** | 0.03 | 0.09 | 0.11 | -0.02 |
| Number of Siblings | -0.02 | 0.22*** | -0.07 | 0.09 | 0.01 | 0.08 | 0.11 | 0.11* |
| War and adversities | 0.55*** | 0.61*** | 0.45*** | 0.54*** | 0.36*** | 0.37*** | 0.29*** | 0.29*** |
| Family violence | 0.10* | 0.36*** | 0.13* | 0.43*** | 0.14** | 0.34*** | 0.16** | 0.37*** |

[a] Full model's adjusted $R^2$ = .37; $F_{(6,325)}$ = 33.86, p < 0.001

[b] Full model's adjusted $R^2$ = .31; $F_{(6,325)}$ = 26.14, p < 0.001

[c] Full model's adjusted $R^2$ = .18; $F_{(6,325)}$ = 12.80, p < 0.001

[d] Full model's adjusted $R^2$ = .14; $F_{(6\ 325)}$ = 10.21, p < 0.001

Note

*p≤.05

**p≤.01

***p≤.001, [e]point-biserial correlation

**Table 3. Results of linear regression models on the different types of child psychopathological symptoms–RC (n = 158).**

| Predictor | PTSD symptoms[a] | | Depression symptoms[b] | | Internalizing problems[c] | | Externalizing problems[d] | |
|---|---|---|---|---|---|---|---|---|
| | Standardized β-Coefficient | zero-order correlation (ρ) | Standardized β-Coefficient | zero-order correlation (ρ) | Standardized β-Coefficient | zero-order correlation (ρ) | Standardized β-Coefficient | zero-order correlation (ρ) |
| Age | 0.10 | .18* | 0.21* | 0.24** | 0.07 | 0.16* | 0.13 | 0.16* |
| Gender (male)[e] | -0.11 | -0.15 | -0.17* | -0.23** | -0.13 | -0.17* | -0.16* | -0.19* |
| Number of Siblings | 0.04 | 0.11 | -0.02 | -0.00 | 0.10 | 0.17 | 0.01 | 0.10 |
| War and adverse events | 0.46*** | 0.47*** | 0.41** | 0.50*** | 0.34*** | 0.35*** | 0.22** | 0.34*** |
| Family violence | 0.20* | 0.38*** | 0.19 | 0.49*** | 0.19* | 0.39*** | 0.21** | 0.39*** |

[a] Full model's adjusted $R^2$ = .35; $F(5,152)$ = 17.63, $p < 0.001$

[b] Full model's adjusted $R^2$ = .35; $F(5,152)$ = 18.12, $p < 0.001$

[c] Full model's adjusted $R^2$ = .22; $F(5,152)$ = 9.83, $p < 0.001$

[d] Full model's adjusted $R^2$ = .17; $F(5,152)$ = 7.30, $p < 0.001$

Note

*$p \leq .05$

**$p \leq .01$

***$p \leq .001$, [e]point-biserial correlation

that resettlement in conflict-affected areas after displacement is increasing the vulnerability for mental health symptoms compared to resettlement in a non-conflict area. The finding that more than half of the children and adolescents (55.4%) reported events of family violence is in line with previous studies with children and families in post-war communities [33, 37] and

**Table 4. Results of linear regression models on the different types of child psychopathological symptoms–IDC (n = 174).**

| Predictor | PTSD symptoms[a] | | Depression symptoms[b] | | Internalizing problems[c] | | Externalizing problems[d] | |
|---|---|---|---|---|---|---|---|---|
| | Standardized β-Coefficient | zero-order correlation (ρ) | Standardized β-Coefficient | zero-order correlation (ρ) | Standardized β-Coefficient | zero-order correlation (ρ) | Standardized β-Coefficient | zero-order correlation (ρ) |
| Age | 0.12* | 0.13 | 0.21** | 0.15* | 0.11 | 0.08 | 0.08 | 0.07 |
| Gender (male)[e] | -0.07 | -0.03 | -0.12 | -0.11 | -0.03 | -0.02 | 0.12 | 0.14 |
| Number of Siblings | -0.05 | 0.09 | -0.11* | -0.01 | -0.03 | 0.05 | 0.12 | 0.15 |
| War and adverse events | 0.58*** | 0.61*** | 0.46*** | 0.50*** | 0.35*** | 0.36*** | 0.28*** | 0.28*** |
| Family violence | 0.06 | 0.25*** | 0.11 | 0.36*** | 0.12 | 0.27*** | 0.17* | 0.32*** |

[a] Full model's adjusted $R^2$ = .35; $F(5,168)$ = 19.19, $p < 0.001$

[b] Full model's adjusted $R^2$ = .26; $F(5,168)$ = 13.15, $p < 0.001$

[c] Full model's adjusted $R^2$ = .13; $F(5,168)$ = 6.30, $p < 0.001$

[d] Full model's adjusted $R^2$ = .14; $F(5,168)$ = 6.84, $p < 0.001$

Note.

*$p \leq .05$

**$p \leq .01$

***$p \leq .001$, [e]point-biserial correlation

supports extant research highlighting that exposure to war trauma is linked to higher levels of family violence experienced by children [31]. In our study, the IDC sub-sample reported significantly higher numbers of traumatic events related to war, displacement and family violence than the RC sub-sample. In line with this, IDC also show higher levels of PTSD and depression.

While the difference in trauma exposure between IDC and RC cannot be explained by the current data, geographical factors not captured in the present study may play a role. For example, some refugee families may have been able to flee their country (Syria) either before or at the beginning of the war and therefore may have experienced less war-related trauma but more challenges related to displacement. On the other hand, the ID families who participated in this study fled from areas in the Salah Al-Din governorate, Iraq, that were deeply affected by complex conflicts, including retaliatory attacks by ISIS and other militias [57]. In addition, our IDC sample included Yazidi children who were exposed to high levels of war and genocide trauma [50].

In line with the above mentioned hypothesis of a transmission of war into family violence [31], greater exposure to traumatic war events may be an important factor in the higher prevalence of family violence among IDC compared to RC. At the same time, post-migration factors related to the legal status of ID families may play a role in the higher rates of both family violence and psychopathology. There is a well-established link between the mental health of displaced people and the ongoing contextual stress of post-migration [41]. In particular, the negative impact of an insecure visa status and temporary residence on the mental health of displaced persons has been scientifically documented in a recent meta-analysis on the mental health of child and adolescent refugees and asylum seekers [11]. As a result of their legal status as internally displaced persons, the ID families in the KRI experience a lack of prospects of permanent resettlement. The distress caused by facing insecure residency may be associated with increased rates of family violence. Although there is no evidence yet on the effect of residence insecurity on parenting behavior, a longitudinal birth cohort study of 4,898 families living in the US [58] found that housing insecurity increased child maltreatment via increases in maternal stress. In addition, the type of housing in our study sample is linked to the legal status of the families; families with IDP status are housed in separate, less permanently planned camps, so there is reason to assume that housing conditions and stressors may also play a role in the higher rates of family violence and psychopathology in the IDC sample. Indeed, previous studies on family violence have shown that a crowded household is a risk factor for both domestic violence [59, 60] and child maltreatment [61–63]. In our study, families with ID status were larger than refugee families and also had to live in more cramped and less secure accommodation (smaller, temporary tents). This difference may be related to the higher levels of family violence within the ID families. In summary, the discrepancy found in war exposure, family violence and psychopathology between the two study groups is consistent with previous research highlighting the particular vulnerability of IDC in war and conflict-affected countries [3, 17].

Multiple linear regression analyses support our hypothesis for the presence of factors contributing to the mental health of war-affected children at different socio-ecological levels. On the individual level, greater war exposure was associated with higher levels of PTSD, depression, and internalizing and externalizing problems. Exposure to war-related and adverse events was the strongest predictor for all mental health outcomes, replicating the dose-response effect of cumulative trauma exposure on the risk of developing PTSD [28, 64–66]. This consistent finding among displaced and non-displaced children and adolescents in different socio-economic and geographical settings is in line with a broad range of literature confirming the detrimental effects of war and conflict on the mental health of minors [3, 8]. In

addition, child age emerged as a risk factor for mental health problems, with older children and adolescents reporting higher levels of mental health problems related to PTSD, depression and externalizing and internalizing behavior problems. This finding is consistent with a small body of evidence suggesting that younger forcibly displaced children have better mental health outcomes than older ones [3], however most evidence on age effects in displaced minors is inconsistent [8]. One possible explanation for the higher levels of psychopathology among older adolescents in our sample could be a lack of future prospects. Gaining emotional and economic independence from parents and choosing a career are major developmental milestones during adolescence [67], and access to education and income inequality are among the strongest determinants of adolescent health worldwide [68]. The lack of educational and employment opportunities in the protracted, socially unstable environment of a refugee camp may have led to a lack of future prospects and negative future expectations, which have been associated with mental ill-health in conflict-affected youth [69]. Female gender was linked to an increased vulnerability to depression whereas no other mental health outcome was significantly influenced by children and adolescents' gender. While greater vulnerability to depression in females, resulting from an interplay of biological, affective, cognitive and sociocultural factors, is a robust phenomenon [70], our findings particularly challenge the well-established assumption of a higher PTSD vulnerability in girls [71]. As the higher risk of exposure to certain types of trauma (sexual assault, child sexual abuse) has been shown to partially account for the differential risk of PTSD between men and women [72], the relatively low number of sexual trauma in our sample may explain why we did not find an association between gender and PTSD.

Applying a socio-ecological perspective to understanding child development after forced displacement, variables in the child's proximal environment, particularly family dynamics, have been shown to be crucial for the mental health of children and adolescents [8]. In the present study, family violence emerged as a significant independent predictor of all psychopathological outcomes, i.e. PTSD, depression, internalizing problems and externalizing problems. This finding is in line with previous evidence on the importance of family-associated factors, which may make an independent contribution to the development and maintenance of psychological disorders in a context of mass or war trauma [19, 32, 33, 37].

Contrary to our expectations, the legal status (ID versus refugee) did not emerge as a significant predictor of any psychopathological outcome when accounting for war- and family violence. However, separate regression analyses for each sub-sample support the notion that different mechanisms may operate in each population within a socio-ecological model of mental health. While on the individual level, the amount of war trauma determined the mental health in both samples, the results at the family level yielded a different picture. In the RC sample, family violence was a predictor of all mental health outcomes except depression, whereas in the IDC, it was only associated with externalizing behavior. There might be several reasons why, in contrast to the RC sample, the IDC did not show a consistent association between family violence and psychopathology. First, given the higher levels of overall trauma, family violence might play a minor role in the IDC sample. In support of this notion, several studies of conflict survivors have found that other types of individual factors, such as gender, may play a subordinate role at high levels of trauma [73, 74]. Secondly, given that the IDC had more siblings compared to the RC, a larger family size might explain the weak link between family violence and children's mental health. In fact, a positive sibling relationship has shown to be a protective factor for children experiencing stressful life events [75]. Moreover, there is evidence supporting a sibling compensation effect in the context of child abuse [76]. It may be assumed that in larger families, the impact of family violence is buffered by siblings taking on responsibilities such as caring for younger children, providing emotional support or shielding

them from harsh parenting, which may lead to a reduction in the impact of parental behavior on minors. Further, within the IDC sample, larger family size was associated with lower levels of depression symptomatology, again suggesting that a larger family size could potentially have a positive impact on the mental health of minors. Despite recent reviews emphasizing the significance of sibling relationships in child and adolescent development and mental health [77, 78], this subject has been understudied, particularly in relation to multiple sibling relationships [77]. Therefore, the results of the present study underscore the need for additional research on sibling relationships and their role in children and adolescents' mental health, aiming to enhance our understanding of both risk and protective factors at the family level.

## Limitations

Since we examined a large sample of children and adolescents by conducting one-on-one interviews including validated and standardized instruments (KID-PIN, [48]), our study provides a valid assessment of risk factors for mental health problems among displaced children in a post-war setting. However, due to the cross-sectional design of the study, our results do not allow a causal interpretation of the associations between the variables. Furthermore, caution is needed regarding the generalizability of the results. Due to the project design, all minors in our study lived with both parents, and thus benefitted from the potentially protective effects of family cohesion [8, 79]. This selection process resulted in the exclusion of at-risk groups, such as unaccompanied displaced minors [80–82], displaced children who experienced family separation [83], or other vulnerable children living in Iraq e.g. those who had lost one or both parents [84] and street children [85].

As a methodological limitation of our study design, it should be noted that our two subsamples, IDC and RC, differ not only in terms of legal status, but also in terms of nationality, a factor directly associated with legal status. However, it cannot be assumed that mere affiliation with a specific nationality is linked to an elevated vulnerability to mental illness. Also, in terms of methodological limitations, a number of factors may have undermined the data collection process. Research in trauma-exposed children and adolescents is necessarily sensitive, so we have taken several precautions regarding participants wellbeing as well as data quality: First, all questionnaires were administered in the form of an interview. This also increased the reliability of the child's reports. However, research has shown that adding mother's reports could have specifically provided a more accurate picture on the experienced family violence [86]. Second, all interviewers received extensive training in psychological interviewing. Still, there may have remained some risk of interviewer bias by collecting data via interviews instead of self-report measures [87].

Finally, another limitation of our study is the lack of assessment of post-migration factors. The relevance of daily stressors in the post-migration environment for mental health has become an increasing focus of research [88, 89]. For instance, lower satisfaction with living conditions in the camp has been associated with depressive symptoms in Syrian adults residing in a camp in Turkey [90, 91]. A study of adult Syrian Kurdish refugees in a camp in the KRI [91] found higher mental health risks as time spent in the refugee camp increased. Especially, as the minors in our sample had already been living in the camp environment for an average of four years at the time of the survey, the impact of the camp-related stressors may be pronounced. Initially designed as a humanitarian emergency setting to meet basic needs, the camp environment has evolved into a permanent residence for the children and adolescents whose childhoods have been shaped not only by war and displacement, but also by the living conditions of a refugee camp.

## Conclusion

The present findings provide important evidence on forcibly displaced minors living in areas of ongoing or recent conflict, constituting the majority of displaced minors worldwide. The high rates of psychopathology observed confirm previous research in conflict-affected Iraq [92, 93] and highlight the urgent need for mental health support for children and adolescents in this region.

Recommending specific interventions may be premature without further replication of our results. However, our findings support the potential benefits of a multi-level approach to addressing mental health needs in affected children and their families. Effective interventions for war-affected children, such as trauma-focused cognitive behavioural therapy [94] and Kid-NET [95], offer potential models for individual-level psychotherapy. Additionally, addressing family dynamics and community-level stressors may be important, but these recommendations should be approached with caution until further studies confirm their efficacy. Future research should also explore additional daily challenges faced by displaced minors and evaluate how post-migration conditions impact mental health. A holistic approach to interventions, incorporating these factors may ultimately prove beneficial.

## Acknowledgments

We are grateful for the involvement of all the families as well as the tireless efforts of all team members in realizing the project. Special thanks to Philippa Specker for editing the manuscript.

## Author Contributions

**Conceptualization:** Jasmin Wittmann, Hawkar Ibrahim, Frank Neuner, Claudia Catani.

**Data curation:** Hawkar Ibrahim.

**Formal analysis:** Jasmin Wittmann.

**Funding acquisition:** Hawkar Ibrahim, Frank Neuner.

**Investigation:** Hawkar Ibrahim, Frank Neuner, Claudia Catani.

**Methodology:** Jasmin Wittmann, Hawkar Ibrahim, Frank Neuner, Claudia Catani.

**Project administration:** Hawkar Ibrahim, Frank Neuner.

**Supervision:** Hawkar Ibrahim, Frank Neuner, Claudia Catani.

**Visualization:** Jasmin Wittmann.

**Writing – original draft:** Jasmin Wittmann.

**Writing – review & editing:** Jasmin Wittmann, Hawkar Ibrahim, Frank Neuner, Claudia Catani.

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
