## [Decision Letter · Decision Letter 0]

26 Jul 2024

PMEN-D-24-00148

Fleeing the war: a socio-ecological perspective on the mental health of internally displaced and refugee children and adolescents living in the Kurdistan Region of Iraq

PLOS Mental Health

Dear Mrs Wittmann,

Thank you for submitting your manuscript to PLOS Mental Health. After careful consideration, I feel that it has merit but does not fully meet PLOS Mental Health’s publication criteria as it currently stands. Therefore, I invite you to submit a revised version of the manuscript that addresses the points raised during the review process.

After asking a number of specialists to review this article, I have only been able to receive a positive response from one of them, who has just submitted his report. You will find this report at the end of the present letter. In order to avoid wasting precious time again finding a second expert, I have taken the initiative of reviewing the research myself.

The reviewer and I particularly emphasise the statistical methodology employed, which deserves to be developed even more rigorously. The reviewer raises a number of important points, which I share, and to which you must respond.

For example, the selection of predictors included in regression models is not always well justified. The decision criteria used to exclude important variables (as those listed in table 1, e.g., level of education) from the regression models are missing. This is a crucial point because some of these variables could potentially increase or decrease the strength of the associations between some of the predictors included in the regression models and the outcomes.

Furthermore, multiple testing is a potential source of false positives that is not taken into account in the current version of the manuscript. The problem of multiple testing could be offset by applying a bootstrapping procedure to the estimation of the parameters of each model (bootstrapped CIs and exact p-values would then be reported in the results).

The reviewer also points out that the discussion considers rather tangential issues (e.g. possible intervention strategies) rather than providing more depth to the critical analysis.

I would also recommend you to significantly improve the quality of the figures as they are not readable.

In any case, both the reviewer and I want you to have the opportunity to be published in Plos Mental Health after revising your work in the light of the major comments made by the reviewer and myself.

I wish you all the best with these revisions and look forward to receiving your revised manuscript.

Kind regards,

Pierre Olivier Jacquet, PhD

Academic Editor

PLOS Mental Health

Reviewers' comments:

Reviewer's Responses to Questions

**Comments to the Author**

1. Does this manuscript meet PLOS Mental Health’s publication criteria? Is the manuscript technically sound, and do the data support the conclusions? The manuscript must describe methodologically and ethically rigorous research with conclusions that are appropriately drawn based on the data presented.

Reviewer #1: Partly

2. Has the statistical analysis been performed appropriately and rigorously?

Reviewer #1: Yes

3. Have the authors made all data underlying the findings in their manuscript fully available (please refer to the Data Availability Statement at the start of the manuscript PDF file)?

Reviewer #1: Yes

4. Is the manuscript presented in an intelligible fashion and written in standard English?

Reviewer #1: Yes

5. Review Comments to the Author

Reviewer #1: 

Thank you for the opportunity to review this manuscript. In the study presented, the authors highlight the different mental health burdens and corresponding predictive factors between refugee and internally displaced children. Although the range of variables interrogated was limited, this remains a valuable and rarely conducted comparison that should be of interest to many scholars. The authors are to be commended for the clear presentation of their study, although I suggest some amendments before publication, as listed below.

Introduction

- The study frames trauma exposure as an individual-level factor which might be perplexing to some readers anticipating a Bronfenbrenner-informed approach. Individual-level factors usually encompass identities, traits, and personal characteristics, whereas trauma exposure (in this case, war exposure and displacement) might be better described as a community-level (exosystem) factor.

- In the final paragraph of the introduction, the authors ask “Are mental health problems higher among IDC ID minors compared to RC, and do the same risk and protective factors contribute to the mental health of forcibly displaced IDC and RC?” I assume “IDC ID” is a typographical error?

Methods

- Although detailed elsewhere, it may be worth reiterating the translation procedure for the study instruments, considering they had to be administered in two different languages.

- In the statistical analysis section, the authors list the predictors chosen for multiple linear regression. The selected predictors, however, are only a subset of the variables reported in Table 1. Was there particular justification for excluding certain variables such as education, religion, ethnicity, place of living etc. from the regression analysis (especially considering the significant difference in some of these variables between IDC and RC)?

- While not mandatory, there may be merit in describing the “1.000 samples bootstrapping procedure” that was used, and why this was favoured over other procedures (e.g. variable transformation).

Results

- If my interpretation of the results is correct, a total of 12 regression models were fitted. Did these need to be corrected for multiple testing?

Discussion

- As part of the study limitations, the authors may wish to comment more on the data collection process (specifically, interviews with minors) and discuss what problems may have undermined this approach, especially given the sensitivity of some of the subject matter (e.g. interviewer bias; limited reliability of child-report etc). Consulting Oh et al., 2018 (https://doi.org/10.1016/j.pedhc.2018.04.021) may be helpful in this regard.

- The authors should exercise caution in interchanging “multiple linear regression” with “multivariate regression” (e.g. see Hidalgo & Goodman, 2013, 10.2105/AJPH.2012.300897). On that note, as an additional sensitivity analysis and a way of reducing the total number of regression models (and risk of multiple testing), the authors might consider a multivariate regression model with all mental health outcomes as dependent variables in a single model.

- At times, the manuscript possibly reaches beyond its reasonable scope. For example, the discussion of intervention strategies may be premature, given that the study did not assess an intervention, and that the study results await replication in other samples.

6. PLOS authors have the option to publish the peer review history of their article (what does this mean?). If published, this will include your full peer review and any attached files.

**Do you want your identity to be public for this peer review?** For information about this choice, including consent withdrawal, please see our Privacy Policy.

Reviewer #1: **Yes: **Andrew K. May

---

## [Editor Report · Decision Letter 1]

18 Dec 2024

PMEN-D-24-00148R1

Fleeing the war: a socio-ecological perspective on the mental health of internally displaced and refugee children and adolescents living in the Kurdistan Region of Iraq

PLOS Mental Health

Dear Dr. Wittmann,

Thank you for submitting your manuscript to PLOS Mental Health. After careful consideration, we feel that it has merit but does not fully meet PLOS Mental Health’s publication criteria as it currently stands. Therefore, we invite you to submit a revised version of the manuscript that addresses the points raised during the review process.

We look forward to receiving your revised manuscript.

Kind regards,

Pierre Olivier Jacquet, PhD

Academic Editor

PLOS Mental Health

Journal Requirements:

Additional Editor Comments (if provided):

Dear authors,

After several unsuccessful attempts to recontact the expert commissioned to evaluate your manuscript in the first round of review so that he could assess your responses to his main comments, I decided to examine your arguments myself.

I think the answers you have provided are satisfactory enough for this article to be accepted for publication in Plos Mental Health.

But before that, I'd like you to add to your revised version an minor element which I think will be useful to your readers: the response you adressed to the Q1) comment I did in the first review round. See below:

______

Editor Q1)

For example, the selection of predictor variables included in regression models is not always well justified. The decision criteria used to exclude important variables (such as those listed in Table 1, e.g. level of education) from the regression models are missing. This is a crucial point as some of these variables could potentially increase or decrease the strength of the associations between some of the predictors included in the regression models and the outcomes.

Authors' response to editor Q1)

Thank you for raising this pertinent point. Our selection of the predictors included in the regression models was based on the existing literature as well as our assessment of the interpretability of the variables based on our impressions at the time of data collection. The following variables were excluded from the regression models for the following reasons:

- Level of education: During data collection, we found that the interpretability of this variable (number of years of education) was limited. As there is no functioning school system in the camps, the variable is more related to factors such as age and gender, rather than reflecting the actual level of education of our sample. We therefore decided not to include it as a predictor in the regression models.

- Religion and ethnicity: On the basis of the existing literature and to the best of our knowledge, it may be difficult to justify the hypothesis that mere religious or ethnic affiliation is associated with higher mental health symptomatology. Due to the lack of scientific evidence and in order not to promote the stigmatisation of religious or ethnic groups, we have refrained from establishing associations between simple group membership and higher mental health symptomatology.

- Place of residence during the war, place of residence before the war: We did not include these variables as they involve retrospective information going back several years for the majority of minors, given that the children had been living in the camp for an average of 4 years, rather than reflecting their current living situation.

______

You can rest assured that I will accept your article as soon as you return the final version of your work.

I wish you all the best.

Pierre O Jacquet
---

## [Editor Report · Decision Letter 2]

15 Jan 2025

Fleeing the war: a socio-ecological perspective on the mental health of internally displaced and refugee children and adolescents living in the Kurdistan Region of Iraq

PMEN-D-24-00148R2

Dear Ms Wittmann,

We are pleased to inform you that your manuscript 'Fleeing the war: a socio-ecological perspective on the mental health of internally displaced and refugee children and adolescents living in the Kurdistan Region of Iraq' has been provisionally accepted for publication in PLOS Mental Health.

Best regards,

Pierre Olivier Jacquet, PhD

Academic Editor

PLOS Mental Health

Dear authors,

I am pleased to announce my decision to accept the manuscript PMEN-D-24-00148R2 for publication in Plos Mental Health.

I would like to thank you for the probity with which you provided the additional elements to avoid over-interpretation of the results. This makes the study more transparent, which can only enhance its quality.

I would also like to thank you for your patience, as the review procedure was, in spite of myself, long and difficult (eg, loss of a reviewer during the procedure).

I wish you all the best.

Pierre O jacquet